

**Relation between the asymmetric ring current effect and the anti-sunward auroral**
**currents, as deduced from CHAMP observations**
Hermann Lühr[1] and Yun-Liang Zhou[2]
1) GFZ, German Research Centre for Geosciences, Section 2.3, Geomagnetism, 14473 Potsdam,
Germany.
2) Department of Space Physics, School of Electronic Information, Wuhan University, 430072
Wuhan, China.

**Abstract.** During magnetically active periods the storm-time disturbance signal on ground
develops commonly an azimuthal asymmetry. Negative deflections of the magnetic horizontal
(H) component are enhanced in the 18:00 local time sector and smallest in the morning sector.
This is commonly attributed to the asymmetric ring current effect. In this study we are
investigating the average characteristics of anti-sunward net currents that are not closing in the
ionosphere. Their intensity is growing proportionally with the amount of solar wind input to the
magnetosphere. There is almost twice as much current flowing in the winter hemisphere as on
the summer side. This seasonal dependence is more pronounced on the dusk than on the dawn
side. Event studies reveal that anti-sunward currents are closely related to the main phase of a
magnetic storm. Since also the asymmetry of storm-time disturbances build up during the main
phase, we suggest a relation between these two phenomena. From a statistical study of ground-
based disturbance levels during magnetically active periods we obtain support for our
suggestion. Observed storm-time disturbance amplitudes are clearly smaller in the summer
hemisphere than in the winter part. This difference increases toward higher latitudes. We propose
a new 3D current system responsible for the zonally asymmetric storm-time disturbance signal
that does not involve the ring current. The high-latitude anti-sunward currents are connected at
their noon and midnight ends to field-aligned currents that lead the currents to the outer
magnetosphere. The net current branch on the morning side is closed along the dawn flank
plasmapause, and the evening side currents along the dusk flank magnetopause. Regardless
through which loop the current is flowing, near-Earth storm-time disturbance level will in both
cases be reduced in the morning sector and enhanced in the evening.


**1. Introduction**
At auroral latitudes intense electric currents are flowing. Due to the anisotropic conductivity
distribution in the ionosphere different current types exist. Quite prominent are the field-aligned


currents (FACs), which can transfer energy and momentum over large distances from the
magnetosphere and deposit it in the high-latitude upper atmosphere. Horizontal Pedersen
currents typically closing these FACs in the ionosphere. Furthermore, there are Hall currents,
flowing perpendicular to the electric and magnetic fields. These are generally regarded as source-
free, and they close in the ionosphere.
The intensity of currents that close FACs in the ionosphere can be estimated from magnetic field
measurements of low-Earth orbit (LEO) satellites on near-polar orbits. By integrating the along-
track magnetic field component over the full orbit the net current flowing transverse to the
orbital plane can be determined. Corresponding results have been obtained from Magsat (e.g.
Suzuki and Fukushima, 1984), Ørsted (Stauning and Primdahl, 2000) and CHAMP (Zhou and
Lühr, 2017) missions. Net currents up to several Mega Ampère (MA) are observed during
magnetically active periods. In principle there are two types of closure currents. Most prominent
are the cross-polar cap Pedersen currents closing excessive Region 1 (R1) FACs, which are not
balanced by R2 FACs. Somewhat weaker are the anti-sunward net currents connecting excessive
downward FACs on the dayside with upward FACs on the nightside. These anti-sunward
currents, carried predominantly by Hall currents, have first been confirmed observationally from
Magsat data (Suzuki and Fukushima, 1982, 1984). Their intensity is clearly controlled by
magnetic activity. Suzuki and Fukushima (1984) suggested that the anti-sunward current is
closed in the magnetosphere through the partial ring current on the duskside. This may be the
cause for the asymmetric storm-time disturbance signal with clear enhancements in the evening
sector.
More recently Zhou and Lühr (2017) provided a detailed study on auroral zone net currents.
Making use of 5 years of high-resolution CHAMP magnetic field data, they could derive the
dependence of theses currents on season, solar wind input and solar flux. In particular, by
deriving current estimates separately for the two hemispheres these dependences emerged very
clearly. The cross-polar cap duskward net current peaks at local summer when the ionospheric
conductivity is high. Conversely, the anti-sunward net current attains largest values during local
winter when conductivity gradients between the auroral region and the polar cap maximise, at
which Hall currents can be diverted into FACs. The out-of-phase variation of these two current
types causes quite different responses of net currents in the two hemispheres to magnetic
activity.
A still open question is the relationship between auroral zone net currents and the asymmetric
storm-time disturbance during the main phase. Suzuki and Fukushima (1984) proposed a closure



of the net anti-sunward current through the duskside partial ring current. It has never been
investigated how the anti-sunward net current flow is split between dawn and dusk side auroral
regions. What is the effect of hemispheric differences in current strength due to seasonal
variation? Can a detailed consideration of all these facts provide hints on the actual 3D geometry
of the net anti-sunward current closure in the magnetosphere?
The C/NOFS satellite on its low-inclination orbit can be used to investigate the ring current
asymmetry. On every revolution it samples ring current signals from all local times. Magnetic
field readings of C/NOFS during the years 2008 through 2010 have been considered by Le et al.
(2011) to study the ring current evolution during storms. The authors show that the disturbance
signal is azimuthally symmetric before and after the storm. But during the main phase a clear
asymmetry is building up, with enhanced amplitudes around the 18 LT sector and reduced values
around 06 LT. During the storm recovery phase, the disturbance signal returns to symmetric
distribution. The degree of asymmetry grows as the magnetic activity gets larger, but the local
time sector in which the largest amplitudes are observed stays around 18 LT. Similar results
concerning the asymmetry of the ring current effect have been derived from ground-based
observations (e.g. Love and Gannon, 2009). These authors claim that the dawn-dusk asymmetry
in the disturbance field is on average proportional to $D_{ST}$. Newell and Gjerloev (2012) made use
of a large number of magnetometers from the SuperMAG data repository. Their SMR index is
similar to $D_{ST}$ but provides local time resolution with four sectors (SMR-00, SMR-06, SMR-12,
SMR-18). By means of a superposed epoch analysis Newell and Gjerloev (2012) determined the
response of their index to a magnetic storm. They found a clear dominance of the disturbance
signal at 18 LT and smallest deflections at 06 LT. All this is consistent with the notion of a
partial ring current on the duskside. For checking that inference Lühr et al. (2017) had a look at
*in situ* ring current density measurements by Cluster and other spacecraft. They could not
confirm the enhancement of ring current intensity in the dusk sector. The strongest ring current
parts are rather observed by these missions in the post-midnight sector. The difference in ring
current interpretation from near-Earth observations and *in situ* measurements has been described
in more details by Lühr et al. (2017), but it is still an open issue.
A quite different explanation for the asymmetric storm-time disturbance signal during the main
phase was suggested by Crooker and Siscoe (1981). They questioned any connection between
the net current and the ring current. According to their analysis the low-latitude magnetic field
effect of the FACs connecting to the net currents at their noon and midnight ends is sufficient to
generate the observed storm-time asymmetry. Interestingly, they made the statements years



before the existence of anti-sunward net currents were observationally confirmed. So far, the
validity of their statements has not been checked in detail.
In this study we make use of CHAMP data and follow up on the results presented by Zhou and
Lühr (2017) for addressing the open questions listed above. Of special interest is the relation
between the net anti-sunward current and the asymmetric storm-time effect at low latitudes.
Prime basis for the investigations is the CHAMP magnetic field dataset from the 5 years, 2001-
2005. But also recordings from geomagnetic observatories are taken into account for
characterizing the near-Earth magnetic effects.
In the sections to follow we will first shortly introduce the data and basic processing algorithms
for determining net currents. Section 3 presents a statistical survey of net currents at all local
times. The dependence of anti-sunward net currents on solar wind input and season is analysed
in Section 4. Section 5 presents for one magnetic storm a direct comparison between anti-
sunward currents and ground-based disturbance levels. The mean characteristics of the ring
current signal during magnetically active periods (Kp > 6), as observed on ground, are outlined
in Section 6. In Section 7 the various observations are discussed, focusing on the comparison
between anti-sunward currents and storm-time disturbance signals. Finally, in Section 8 results
are summarised and a new 3D current system is proposed for closing the anti-sunward net
currents.

**2. Dataset and calculation of net auroral currents**
The CHAMP satellite was launched into a near-circular polar orbit (inclination: 87.3º) with an
initial altitude of 456 km on 15 July 2000 [Reigber et al., 2002]. By the end of the mission, 19
September 2010, the orbit had decayed to 250 km. The orbital plane covers all local times within
130 days when considering upleg and downleg arcs. The Fluxgate Magnetometer (FGM) on
board CHAMP recorded the vector magnetic field every 0.02 s with a resolution of 0.1 nT. The
FGM magnetic field readings are calibrated routinely by using the observations of the onboard
absolute scalar Overhauser Magnetometer. In this study the fully calibrated Level-3 magnetic
field products (product identifier: CH-ME-3-MAG) are used (Rother and Michaelis, 2019),
which are provided in the North-East-Center (NEC) frame with a time resolution of 1 Hz. The
time period used in this study comprises the five years from 2001 to 2005, experiencing solar
and magnetic activities from high to moderate levels. Five years of CHAMP magnetic field
observations are just needed to sample all local times 14 times, evenly distributed over all
seasons.





The approach for deriving net currents in the auroral region from CHAMP magnetic field data
has been described in detail by Zhou and Lühr (2017). Here we use the same dataset and adopt
their processing algorithm. Calculations are based on Ampère's law in integral form
$$I = \frac{1}{\mu_0} \oint_L B_{AT} dl \qquad (1)$$

where $I$ is the net current flowing through the closed integration contour, $\mu_0$ is the permeability
of the free space, $B_{AT}$ is the along-track magnetic field component caused by the current $I$, $dl$ is
a differential path element along the CHAMP orbit. Equation (1) can be written in discrete form
as
$$I = \frac{1}{\mu_0} \sum_{m=1}^{n} B_{AT\,m} \cdot \varDelta l \qquad (2)$$

where $m$ is the summation index, and $\varDelta l$ is the path length per increment (here 7.56 km for 1s).
For deriving the along-track magnetic field component, $B_{AT}$, we have subtracted from the
CHAMP data the main field, crustal field and large-scale magnetospheric field, as represented
by the high-resolution model POMME-6 (Maus et al., 2010). From the set of magnetic
residuals, the component aligned with the velocity vector is calculated.
Zhou and Lühr (2017) derived net currents from integration along full CHAMP orbits. In
addition, they applied integration loops confined to one hemisphere and could study
hemispheric differences. Here we go one step further by estimating net currents flowing through
a loop from subauroral latitudes up to the geomagnetic pole. In this way we get current estimates
for all local times and can compare intensities on the dawnside with those on the duskside and
noon with midnight results. The penalty for the further detailing of the results is that we have
to make certain assumptions on the magnetic fields along parts of the integration path where no
direct observations are available. The considered integration paths for the two local time sectors
along the orbit are sketched in Figure 1. CHAMP magnetic field readings are taken from 50°
magnetic latitude (MLat) (point A) up to the highest MLat reached along the orbit (point B).
From there the virtual return path goes vertically down to point C, follows the Earth's surface
until point D and goes vertically up to the start point A. The second loop follows the same
scheme, taking CHAMP readings along the track from E to F and closing the path along the
virtual track (F-G-H-E).
Since there are no measurements along the return path, we have to make assumptions about the
magnetic field along that track. Here we follow the same reasoning and approach as



successfully applied in the work of Zhou and Lühr (2017). Auroral net currents are connected
to FACs on both ends. According to Fukushima's theorem (Fukushima, 1976) magnetic
signatures from a pair of antiparallel FACs closed by ionospheric currents vanish at the Earth's
surface. The current configuration in our case, however, differs somewhat from the ideal case
presented by Fukushima (1976), therefore the theorem might not be fully applicable here.
For estimating the contributions from the unsampled parts the following assumptions are made:
(1) The contribution from C→D is similar in shape to that from A→B. (2) The contribution
from D→A are proportional to the vertical field component $Bz$ at point A since the radial
magnetic field varies only smoothly through the current sheet. An outcome of this exercise is
that the integral over A→B has to be multiplied by 1.2 for including the contributions from
path C→D and that the vertical magnetic field component, $Bz$, has to be multiplied by 11 times
the orbital altitude and divided by the permeability of free space to represent the contributions
from path D→A. For further justification of these corrections see Zhou and Lühr (2017),
Section 4.2. The same approach described here is also applied to the contour E-F-G-H-E. The
remaining paths in the integration loops are B-C and E-H. Here again, the observed $Bz$
component at the points B (E) have been taken as a measure for scaling the missing
contribution. We have tested a series of different factors multiplied to the $Bz$ values at the top-
side corners. There is a way to validate the suitability of the applied factors. Each local time
sector is sampled in two ways, on upleg and 130 days later on downleg passes. In these two
groups the ring integral is calculated in opposite directions. Only in the case of a proper scaling
of this vertical contribution, both results are identical. From this test we found that the best
agreement is obtained when the contributions from the vertical path elements in the middle are
neglected. Figure 2 shows the final comparison for both hemispheres and all local times. For
the northern hemisphere (left frame) we obtain in this way an almost perfect match between
upleg and downleg results. The agreement is not as good for the southern hemisphere, but any
additional contribution from these vertical paths makes the agreement between the curves
worse.

**3. Statistical survey of net current distribution**
For obtaining the average distribution of net currents at all local times we consider CHAMP
magnetic field data from the 5 years, 2001-2005. Overall 24,440 orbits with clean data are
available. From each orbit we obtain two net current results for both hemispheres. This results
in a large number ($\sim 10^5$) of samples for this study. Figure 3 shows the average local time





variations of net currents in the northern and southern hemispheres (upleg and downleg results
are combined). Positive values represent eastward currents. On average we find somewhat
larger values in the northern hemisphere than in the southern. This is consistent with the
observations of Zhou and Lühr (2017). Positive (eastward) net currents prevail within the local
time sector 07-19 MLT, representing a dawn to dusk flow. The opposite sign is found in the
other 12-hour local time sector, reflecting also dominant dawn to dusk currents.
There is not only a local time variation of the net currents but also a dependence on season.
Figure 4 shows the distribution of current strength in a magnetic local time (MLT) versus Month
of Year frame. We clearly find strongest currents during local summer months in particular
around noon at both hemispheres. This is primarily due, as explained by Zhou and Lühr (2017),
to the enhanced ionospheric conductivity during that season.
Another quantity that is expected to influence the net current, is the orientation of the
interplanetary magnetic field (IMF). Here we have checked the dependence on the IMF $By$
component. As can be deduced from Figure 5, there is some dependence on the sign of IMF
$By$. For positive $By$ clearly stronger net currents are observed in the northern hemisphere around
the noon sector during months around June solstice. Our explanation for the difference in
eastward net current is the DPY effect (e.g. Wilhjelm et al., 1978; Clauer et al., 1995). The
direction of this daytime auroral current depends on the sign of IMF $By$. For negative $By$ it
flows in opposite direction to the dawn to dusk currents in the polar cap. The related effect is
also expected in the southern hemisphere but for negative $By$ around December solstice. Here
it is not so obvious. It seems that the period with positive IMF $By$ were more active and thus
masked the IMF $By$ effect in the southern hemisphere.
As outlined by Zhou and Lühr (2017), the large net currents derived from noon/midnight
orbits can be related to the cross-polar cap Pedersen currents closing the excessive Region 1
(R1) FACs. The positive values around noon and the negative around midnight are both
consistent with that notion. In this study we are more interested in the net currents on the
dawn and dusk sides. Therefore, we consider the average values within the local time sectors
03-09 MLT and 15-21 MLT as dawnside and duskside net currents, respectively. From Figure
3 it is evident that a negative (westward) average current results from the 03-09 MLT sector
and a positive (eastward) from the 15-21 MLT sector. This means, both sides contribute to an
anti-sunward net current. The characteristics of these anti-sunward currents are of prime
interest for this study.

**4. Dependence of net current on solar wind input and on season**
Similar to Zhou and Lühr (2017) we also investigate the dependence of anti-sunward net
currents on magnetic activity. Different from them we look at the fractions flowing on the
dawn and dusk sides separately. As measure for the solar wind input, we use the coupling
function as defined by Newell et al. (2007). By somewhat rescaling this function we obtain
the so-called merging electric field, $E_m$, which represents approximately the solar wind
electric field in units of mV/m
$$E_m = \frac{1}{3000} V_{SW}^{\frac{4}{3}} (\sqrt{B_y^2 + B_z^2})^{\frac{2}{3}} \sin^{\frac{8}{3}}(\frac{\theta}{2})$$    (3)
where $V_{SW}$ is the solar wind velocity in km/s, $B_y$ and $B_z$ both in nT are the IMF components
in GSM coordinates, $\theta$ is the clock angle of the IMF. $E_m$ values have been smoothed over 15
min, and the propagation time from the bow shock to the ionosphere has been considered by a
delay of 20 min (for more details see Zhou and Lühr, 2017).
Figure 6 shows the mean dependence of the eastward net currents on the dawn and dusk sides
on the merging electric field, $E_m$, separately for the northern and southern hemispheres. The
current values had been grouped into five activity classes ($0<E_m\leq1$, $1<E_m\leq2$, $2<E_m\leq3.5$,
$3.5<E_m\leq5$, $5<E_m\leq7$ mV/m). Blue dots represent the mean values within these classes and the
blue bars reflect the standard deviations. The mean values infer a good linear relationship
between current intensity and merging electric field in all cases, as confirmed by the fitted red
lines. On the dawnside westward currents get stronger with growing $E_m$ and correspondingly
eastward currents intensify on the duskside. This confirms in all four cases an increase of anti-
sunward currents with growing activity. Slopes are somewhat steeper on the dawnside than on
the duskside. Interestingly, the net currents on the dawnside show a small positive bias (~52
kA) for vanishing solar wind input. We relate that to the effect of net anti-sunward plasma
flows driven by intense day-to-night winds in the early morning sector (e.g. Lühr et al., 2007)
during quiet periods.
As expected, the net currents on the flanks depend also on season. Figure 7 shows the annual
variation of eastward net currents on the dawn and dusk sides separately for the two
hemispheres. This analysis is based on data from more active periods with $E_m > 3$ mV/m
(approximately $Kp > 4^+$) since anti-sunward net currents are phenomena increasing with
magnetic activity. We find in both hemispheres weaker anti-sunward currents at local summer
than at local winter. This holds for the dawn and dusk sides and is consistent with the results
of Zhou and Lühr (2017). Compared to the mean values, the relative annual variations are not



too large (15% - 20%) and have comparable sizes in both hemispheres. In the northern
hemisphere a semi-annual signature is quite prominent, commonly referred to as the Russel-
McPherron effect (Russel and McPherron, 1973). It reflects the typical annual variation of
magnetic activity with maxima at equinoxes and a minimum around June solstice. The semi-
annual variation is not so obvious in the southern hemisphere, but the annual amplitude is
larger.
For completeness we have also calculated the dependence of the dawn and dusk side net
eastward currents on solar wind input separately for June and December solstice months and
for the two hemispheres. Obtained results are listed in Table 1. The negative signs of the
slopes on the dawnside and the positive on the duskside represent both increasing anti-
sunward current intensity with enhanced solar wind input. When comparing the slopes of the
dawn and dusk sides between the two solstices, one finds a smaller seasonal difference on the
dawnside than on the duskside. Here the factor is partly reduced to less than a half during
local summer with respect to local winter. Net currents in the dusk sector are obviously less
dependent on solar wind input during times of a sunlit ionosphere. This is surprising since
Guo et al. (2014) reports that the eastward auroral electrojet intensity shows a larger seasonal
variation (stronger in local summer) than the westward jet. Obviously, the more pronounced
conductivity gradients between the auroral region and the polar cap during dark seasons play
a larger role for net currents. Finally, it is interesting to note that in Table 1 the intersects on
the dawnside show systematically large sunward net currents (82 kA) in the summer
hemispheres. This is consistent with the stronger day-to-night wind in the sunlit polar region
(e.g. Lühr et al., 2007) which seem control the anti-sunward plasma flow over the dawnside
polar cap during quiet times.

**5. Variation of net currents during a magnetic storm**
It is suggested since quite some time that the anti-sunward currents are connected via FACs to
the ring current (e.g. Suzuki et al., 1985). In particular, it is believed that net currents feed the
partial ring current on the duskside. Here we want to check, to which degree the CHAMP data
support this inference. The partial ring current generally forms during the main phase of a
magnetic storm.
For investigating these connections in more details, we have selected the geomagnetic storm
on 17 August 2003. This event is well suited because CHAMP is crossing the auroral oval on
orbits close to dawn/dusk. The storm is initiated by a sudden storm commencement (SSC) at

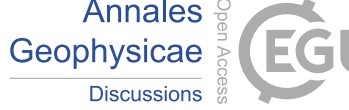

14:20 UT on 17 August. From the solar wind and interplanetary magnetic field (IMF)
variations, shown in Figure 8 (bottom), we can deduce that a sudden increase of solar wind
speed from about 420 km/s to more than 500 km/s is responsible for the SSC. About an hour
later, when IMF $Bz$ turns negative, the main phase of the storm starts and extends into the
next day. On that day the storm time disturbance index reached a minimum of $D_{ST}$ = -148 nT
(see Fig. 8, top frame). It follows a typical recovery phase lasting several days. During part of
that time IMF $Bz$ is still negative, but the solar wind speed has returned to pre-event levels.
For comparison we present in the top frame of Figure 8 the storm-time evolutions of the total
anti-sunward net currents (blue curves) including contributions from both hemispheres
together with the SYM-H index (red curves). The SYM-H values are averages over the 10-
min intervals when CHAMP crossed the polar regions. Right after the southward turning of
IMF $Bz$ intense anti-sunward currents (negative values) commence. About 4 hours later
currents recover to a moderate value, but intensify again early next morning. This intermittent
occurrence of net current continues into the recovery phase of the storm but with decreasing
amplitudes.
So far, we have seen the evolution of total net current intensity during the magnetic storm on
17 August. More details can be derived from Figure 9, where the contributions from the two
hemispheres are shown separately. The current signatures are quite different in the four
sectors. Before the SSC net currents in all frames are close to zero. Particularly intense anti-
sunward currents, up to 2 MA, appear in the southern hemisphere (SH) on the dawnside
during the main phase. Some hours before this strong signal, less intense anti-sunward
currents are observed on the dawnside in the northern hemisphere (NH) and the duskside SH.
It is interesting to note that there is in general a synchronous variation of net currents in these
two antipodal sectors with somewhat smaller amplitudes in the south. For example, the
prominent negative peaks around 42h Event Time (ET) in both hemispheres, which occur at
the start of the recovery phase. Even later in the recovery phase (~55h ET) a sizable anti-
sunward current appears in the SH dawn sector. Different to the other sectors there is only
little net current activity on the NH duskside. Quite common for all four sectors, there is
hardly any net current activity during times of northward IMF.
For the interpretation of the observations we have to remind that the event takes place towards
the end of northern summer. More intense anti-sunward currents are therefore expected in the
SH. Also the quietness on the NH duskside is consistent with our previously shown statistical
results for that season. The quasi-synchronous variation of net currents at NH dawn and SH



dusk could convincingly be explained with a control by IMF *By* on related FACs in the polar
cap. Stronger anti-sunward currents are expected in the NH dawnside for negative IMF *By*
and in the SH dawnside for positive IMF *By*. A direct comparison with the IMF observations
reveals a qualitative agreement. For example, the intense SH dawn current matches well the
positive excursion of IMF *By* around 30h ET, but the details of phasing do not fit so well in
other cases. At least for this event we can state that in both hemispheres more intense anti-
sunward net currents are observed on the dawnside than on the duskside.
It would have been desirable to study more individual storms in this detail. But an event has
to satisfy a number of conditions for providing instructive results on the temporal evolution of
anti-sunward currents during a storm. The storm should occur close to one of the solstice
seasons, and the local time of the CHAMP orbit has to be close to dawn/dusk. We have
considered all storms during the CHAMP era (2000-2010) with $D_{ST}$ exceeding -100 nT. Just
the presented event satisfied all these requirements reasonably well.

**6. Ground-based signature related to anti-sunward net current**
The observed anti-sunward currents are connected on both ends to FACs. These field-aligned
currents have to close somewhere in the magnetosphere. Depending on the route these
currents take corresponding magnetic signatures are expected at Earth surface. Traditionally
the $D_{ST}$ index (or SYM-H, as shown in Figs. 8 and 9) is used for describing the evolution of a
storm. But this index reflects only the azimuthally symmetrical part of the magnetospheric
fields. Therefore, it is not well suited to reflect the asymmetric effects possibly caused by the
auroral net currents. More appropriate for this purpose seems to be the SuperMAG ring
current index, SMR. It is a quantity comparable to $D_{ST}$ or SYM-H but provides local time
resolution from four sectors (SMR-00, SMR-06, SMR-12, SMR-18). More details about the
SMR index can be found in Newell and Gjerloev (2012). By comparing the evolution of
magnetic signatures on the evening and morning sides (SMR-18 and SMR-06) we may see
the effect of a partial ring current. Figure 10 shows in the top frame the field deflections in
these two time sectors during our storm. As expected, there are larger amplitudes observed on
the evening side, in particular towards the end of the main phase. In the lower frame the
differences between the two traces, SMR-18 minus SMR-06, are plotted. In this way we try to
eliminate the contribution of the symmetrical ring current. Before and after the active phase of
the storm the difference stays close to zero. Shortly after the SSC we find first positive
deflections, i.e. a dominance of the dawn sector, and at the end of the prominent minimum,



i.e. larger effects on the duskside. Thereafter the difference signal is more variable. A closer
comparison between SMR difference signal and the net currents in Figure 9 reveals that the
best (but not perfect) match is found with the SH dawnside currents. However, these would,
according to the traditional picture weaken the ring current on the morning sector. At least for
this storm the asymmetric $D_{ST}$ effect cannot be explained by an intensification of the duskside
ring currents. We will revisit this issue in the discussion, Section 7.
From our study of the anti-sunward net currents we know that the effects are significantly
different in the two hemispheres mainly depending on the season. Although SMR provides
information on local time differences, it does not distinguish between hemispheric sources.
In order to obtain more information on the net current seasonal effects in ground observations
we analysed magnetic field data from a meridional chain of observatories. Stations involved
are Wingst (WNG, 54.15° DLat), L'Aquila (AQU, 42.45° DLat), Tamanrasset (TAM, 24.80°
DLat), Bangui (BNG, 4.36° DLat), and Hermanus (HER, -33.86° DLat), where DLat is the
latitude in dipole coordinates. Our study has shown that net currents are particularly strong
during magnetic storms. We are therefore interested in magnetic field deflections at the
observatories during disturbed times. The disturbance signal is determined from times with a
magnetic activity index Kp >= 6. Here the values around 06 and 18 MLT are considered since
they are expected to show the largest difference. For obtaining them we took the hourly
averages of the horizontal component, H, from 04+05 UT and 16+17 UT, respectively. A
quiet-time background field is subtracted, determined from hourly averages of the same UT
times as above but only data within the Kp = 0-1 range are selected. In order to make the
result well comparable with our net currents we considered the same 5 years (2001-2005) as
for CHAMP.
The obtained mean horizontal disturbance fields are listed in Table 2 separately for the three
Lloyd seasons: June solstice (May-Aug), December solstice (Nov-Feb) and combined
equinoxes (Mar+Apr, Sep+Oct). As expected, we get negative mean values (southward fields)
in all the cases. The values in the evening sector are more negative than those from the
morning sector. An exception makes the station WNG. Here all the dusk fields, opposed to
the other observatories, are more positive than those from dawn. This observatory is located
obviously too far north. Therefore, its readings are affected also by the auroral electrojet
during severe storms, not only by the ring current. For that reason, we have not considered it
any further in the analysis. The larger amplitudes at dusk than on the dawn side are
traditionally attributed to the effect of the partial ring current. Overall means of $H_{dawn} - H_{dusk}$


are: BNG: 83 nT, TAM: 73 nT, HER: 57 nT, AQU: 47 nT. The differences decrease with the
distance from the geomagnetic equator.
The storm-time disturbance fields at the observatories vary from season to season. Largest
values are obtained for the months around December and smallest around June. This reflects
the distribution of strong storms during the 5 years considered. In order to find the relative
level of disturbance at a given observatory, values have to be compared within a season. For
this purpose, we selected BNG as the reference station because it is located close to the
geomagnetic equator. Table 3 lists the ratios of the various stations separately for the seasons
and the dawn and dusk sides. In principle, the expected trend emerges, observatories at higher
latitudes record smaller disturbances. We have to note, however, that the data from BNG are
partly quite disturbed. It seems, some instrumental problems have existed during those years,
due to political unrest. This is in particular true for the dawn values. We tried to eliminate bad
values from the sample. But still, the dawn references seem to be either a little low (equinox)
or too high (June). Of interest is the relative decrease of the values with increasing latitude.
When starting with the equinox seasons in Table 3 we find the expected order. This is not true
for December solstice months. Here the highest latitude station AQU exhibits a larger ratio
than HER. Conversely around June solstice, here the ratio at HER is much larger than at AQU
and close to that at TAM.
For the interpretation of these ratio variations we like to recall the seasonal, hemispheric
variations of the anti-sunward net currents, as shown in Figure 7. In both hemispheres the
current strength peaks during local winter. This characteristic is also observed on ground;
observatories in the winter hemisphere experience larger storm-time disturbance levels. The
auroral field-aligned current systems feeding the anti-sunward net currents contribute also to
mid-latitude magnetic signatures primarily in the same hemisphere.
The $D_{ST}$ values are commonly interpreted as caused by a ring current in the equatorial plane.
Therefore, a reduction of the deflection level with latitude is expected, proportional to *cos ß*,
where *ß* is the magnetic dipole latitude. In Table 4 we compare the derived ratios of H
deflections with the expected cosine values for our four stations. In this case the results are
ordered by local summer and winter. The observed ratios in the winter hemisphere follow
reasonably well the cosine law. Conversely, the summer results fall progressively short
towards higher latitudes. Note, for June solstice, dawnside, we have deliberately used a 10%
smaller reference than the BNG readings (see arguments above) to make the ratios of the





other three observatories more consistent. In the next section we will offer some suggestions
for the reduced summer-time disturbance levels.
The ring current signal has also been measured by the C/NOFS satellite. On its low-latitude
orbit (inclination: 13°) it samples the H component deflections at all local times on every
orbit. In that way, any azimuthal asymmetries of the signal can well be detected. In a
dedicated study, Le et al. (2011) investigated the evolution of the ring current signals during
several geomagnetic storms. They clearly could confirm the appearance of an asymmetry
during the storm main phase. During the recovery phase the signal became symmetric again.
In a later study Lühr et al. (2017) performed a statistical survey on the type of asymmetry. For
different classes of magnetic activity, the mean difference between dawn and dusk deflections
were determined and the local time where the maximum appeared. For high activity, $Kp > 6$,
they obtained a center displacement of 38 nT, half the difference between dawn and dusk
signals. This can be compared with the difference of disturbance levels we derived here for
periods of $Kp > 6$ from the observatories. For BNG, closest to the equator, we got a mean
value of 83 nT, which is slightly more than the corresponding result from C/NOFS (76 nT).
The explanation for the difference between the two values could be the occurrence of larger
storms during our analysis period (2001-2005) as compared to the years from 2009 through
2013 considered for C/NOFS statistics. Interestingly, C/NOFS finds for all levels of activity
largest ring current signals on average near 18 MLT. This suggests a large-scale 3D current
system connected to the anti-sunward currents rather fixed in local time.

**7. Discussion**
In this study we investigated the statistical properties of anti-sunward net currents in the
auroral regions and their relation to ground-based signatures at middle and low latitudes. The
general properties of auroral net currents had been presented by Zhou and Lühr (2017). Here
we go one step further by determining the anti-sunward currents flowing on the dawn and
dusk sides separately. As expected, the current intensity is directly proportional to the solar
wind coupling function, $E_m$. When looking at annual averages the resulting net currents are
about the same for enhanced activity (e.g. $E_m > 3$ mV/m) in the dawn and dusk sectors and in
both hemispheres (see Fig. 7). However, obvious differences appear when taking the local
seasons into account. From Table 1 we can deduce that the slopes of the current intensity
curves with respect to $E_m$ are similar on the dawnsides for local summer and local winter.
Conversely on the duskside, the obtained $E_m$ dependences are clearly steeper for winter than



463 for summer conditions. This is valid for both hemispheres. We interpret it as an indication

464 that the conductivity gradient on the duskside between the auroral region and the polar cap is

465 much steeper in the winter hemisphere than in the sunlit summer. Different from that

466 dawnside conductivity gradients seem to be less season dependent.

467 When evaluating the average hemispheric net current characteristics from Table 1 we obtain

468 for $E_m = 6$ mV/m ($Kp \approx 6^+$) intensities of about 640 kA and 810 kA in each hemisphere for

469 summer and winter conditions, respectively. It has been reported earlier (e.g. Guo et al., 2014)

470 that the intensity of the eastward electrojet on the duskside is depending more directly on the

471 sun-induced conductivity. But we find, the stronger summer-time eastward electrojet

472 obviously contributes less to the anti-sunward net currents. The closure of the electrojet

473 currents across the polar cap seems to be quite efficient during the sunlit season.

474 A detail, interesting to note, is that for vanishing solar wind input, $E_m = 0$, i.e. due northward

475 IMF, we obtain, in particular on the dawnside during summer season, sunward net currents of

476 about 80 kA in both hemispheres. Reason for that is probably the day-to-night wind over the

477 polar cap that is driving net currents in opposite direction. More dedicated studies would be

478 needed for elucidating the details of a high-latitude wind dynamo under such special

479 conditions.

480

481 7.1 Comparison with ground-based observations

482 We have shown that the magnetic field effects of anti-sunward currents are also observable on

483 ground. Our statistical study of recordings along a European-African meridional chain

484 revealed that the high-latitude seasonal differences are visible on ground. The asymmetry

485 between dawn and dusk disturbance signals during magnetically active periods is larger in the

486 winter hemisphere, and the seasonal effect becomes more prominent at stations on higher

487 latitudes. This strongly suggests that the full 3D current circuit, connected to the anti-sunward

488 currents in each hemisphere, is contributing mainly to the mid-latitude magnetic signal in the

489 same hemisphere. This means, recordings in the summer hemisphere underestimate the $D_{ST}$

490 value. The over-proportional reduction of the mean $D_{ST}$ index during months around June

491 solstice, compared to other activity indices, e.g. Kp, has earlier been reported (e.g. Mursula

492 and Karinen, 2005). In their Figure1 they show that the average H component deflection value

493 at the northern hemisphere index observatories reaches almost 0 nT at the beginning of July.

494 While at Hermanus the zero level is attained around new year. In our view the $D_{ST}$ minimum

495 can be explained by the combined effect of the well-known annual July magnetic activity

496 minimum with the weaker disturbance signal in the summer hemisphere. Since three out of

497 four $D_{ST}$ observatories are located in the northern hemisphere, the excessive July minimum in

498 $D_{ST}$ is expected as resulting of a hemispheric bias. Just for completeness we may note that

499 Mursula and Karinen (2005) offered another explanation for the $D_{ST}$ July minimum that we

500 do not regard as so convincing.

501 Rather interesting features are revealed from the event study of the magnetic storm on 17

502 August 2003. The evolution of sunward currents, as shown in Figure 9, is quite different on

503 the dawn and dusk sides in the two hemispheres. Several of the statistical features presented

504 in the previous sections can also be found in this event that occurred during northern summer

505 conditions. Largest currents are detected in the southern, winter hemisphere on the dawnside

506 during the storm main phase. In the northern summer hemisphere the duskside currents

507 exhibit only small amplitudes. This is consistent with the mean seasonal dependences of this

508 local time sector (see Table 1). Sizable net currents appear on the dawnside in the northern

509 hemisphere at times when they are low in the southern hemisphere. This hemispheric

510 alternation in current flow can be related to the varying direction of the IMF *By* component.

511 For checking the magnetic effects of the net currents on ground we had a look at the SMR

512 index for this event (see Fig. 10). We expected a clear dominance of SMR-18 over SMR-06.

513 But only a moderate negative difference appears towards the end of the main phase in the

514 lower frame of that figure. Over large parts of the storm-time the signal is varying about the

515 zero-line. For the interpretation of this result we have to note that most of the observatories

516 contributing to the SMR index are located in the northern hemisphere. Because of the

517 prevailing summer season, the ring current index is expected to be underestimated (see Table

518 4). This can be regarded as a general problem. Also for the determination of the $D_{ST}$ value,

519 three observatories in the northern hemisphere are used and one in the southern. This leads to

520 a systematic underestimation of the ring current index during the months around June solstice

521 compared to those from December months.

522 There is a certain anti-phase variation of the SMR difference in Figure 10 with the sunward

523 currents in Figure 9 on the NH dawnside and SH duskside. Prominent peaks appear around

524 19h and 41h ET in both figures but with opposite sign. This indicates that at the listed peak

525 times the negative deflections in the northern hemisphere are stronger on the dawnside than

526 on the duskside. The largest negative peak in the SMR difference signal, around 30h ET, is

527 well aligned with the strong anti-sunward current on the SH dawnside, but it is not as large as

528 expected from the strong net current in the SH. This observation provides clear evidence that

 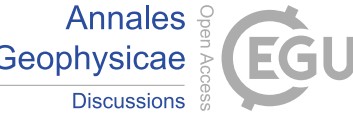

the effect of auroral net currents can directly be recognised by mid-latitude observatories, but
the two hemispheres should be interpreted separately. With the present distribution of
stations, contributing to SMR, however, it is expected that the asymmetry of the ring current
effect is underestimated during the season around June solstice, like the presented case, when
stronger anti-sunward currents flow in the southern hemisphere.

7.2 Suggestion for a 3D current circuit
When comparing the CHAMP net currents at the four quadrants with the temporal evolution
of the SYM-H or SMR indices we find strongest net currents in the dawn sector and
particularly in the southern, winter hemisphere (see Fig. 9) during the storm main phase. The
traditional suggestion was that the auroral net currents, in particular those from the evening
sector, are connected to the ring current and intensify the part in the dusk sector (e.g. Suzuki
et al. 1985). But just on the duskside we find only week anti-sunward currents during our
August 2003 storm. In previous works the term "partial ring current effect" is frequently used.
This was mainly meant as an acronym for an azimuthally asymmetric disturbance signal
during magnetic storms. The presented observations in this paper and previous publications
considering *in situ* ring current density distributions (see Lühr et al., 2017 for a review)
provide little evidence for a direct connection between auroral net currents and the ring
current. Here we want to introduce our idea of the 3D current circuit connected with the anti-
sunward currents.
From electrodynamic considerations it can be assumed that the FACs connected to the net
currents appear at steep conductivity gradients. This locates them at fairly high latitudes near
the border between auroral oval and polar cap. Field lines from this border do not connect to
the ring current but reach out close to the magnetopause. During the storm main phase a lot of
current flows along the electrojets from the day to the night side, which cannot be returned to
the dayside across the poorly conducting polar cap (in particular in the dark hemisphere). The
excessive current flows out along field lines to the magnetopause on the dawn and dusk side
flanks. Here it is conducted back to the dayside. Figure 11 presents a schematic drawing of
the envisaged 3D current circuit. Shown is a view onto the northern hemisphere. Equivalent
current routes are assumed on the southern side. There is no connection to the ring current
foreseen.
A current through our dawnside circuit will generate a northward magnetic field on ground,
thus reducing the $D_{ST}$ effect. Conversely, net currents through the dusk loop cause a

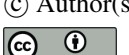



southward field enhancing the ring current effect. Regardless on which side the net currents
close the same kind of asymmetry results. The near-Earth disturbance signals from these
current circuits are dominated by the magnetic effects of the connecting FACs. For the
resulting signals, it does not make a big difference whether the currents close through the ring
current or further out in the magnetopause. Support for our current model comes from
Haaland and Gjerløv (2013), who report, based on Cluster observations, enhanced sunward
magnetopause currents on the dusk flank during the main phase of a storm. With the 3D
current circuit suggested here, it makes no problem to understand, why enhanced disturbance
levels always appear around 18 MLT (see Le et al., 2011) independent of the magnetic
activity level. Already Love and Gannon (2009) had noticed that storm-time disturbances are
commonly higher around the 18 MLT sector. They even suggested a linear relation between
the asymmetry amplitude and the $D_{ST}$ value. The asymmetry should amount on average to
about 20% of the $D_{ST}$ value. This claim was challenged by Siscoe et al. (2012). These authors
tried to identify a magnetospheric process that could systematically enhance the ring current
intensity in the dusk sector. In the end they were not able to offer a convincing explanation.
We claim that our 3D current circuit, driven by the high-latitude net currents, can better
explain the observed features of the asymmetry signal. It seems to be quite stable in space.
Therefore, the localisation to 18 MLT is achievable. We do not believe in a dependence of the
asymmetry amplitude on the $D_{ST}$ value. The anti-sunward currents are closely controlled by
the solar wind input ($E_m$ value). But in a statistical sense, $E_m$ and $D_{ST}$ are related, therefore the
result of Love and Gannon (2009) can be understood. We claim, there is no direct connection
between the ring current activity and the asymmetric storm-time signal. More studies of
magnetic fields and currents in the outer magnetosphere are needed to confirm our 3D current
configuration.

**8. Summary and Conclusions**
In this study we have investigated the auroral net currents flowing anti-sunward. For the first
time, we present the partitioning of contributions from the dawn and dusk sides and from the
two hemispheres to the total net current. These magnetic storm-time phenomena show
significant dependences on solar wind input, season, and IMF *By* orientation. Of particular
interest here is the complete current circuit including the field-aligned currents attached to the
anti-sunward currents and the closure in the magnetosphere. Important results may be
summarised as follows:

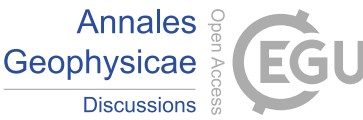

1. Anti-sunward currents grow on average proportionally with the solar wind input (merging electric field, $E_m$). This is valid for the dawn and dusk sides and for all seasons.

2. More intense currents are observed in the winter than in the summer hemisphere. We relate that to the steeper conductivity gradients between auroral zone and polar cap during dark seasons. Then a larger part of the electrojet return current has to be by-passed through the magnetosphere via FACs.

3. On average, more anti-sunward current is flowing on the dawnside (10%-20%). The seasonal dependence of net currents is larger on the duskside. In the sunlit summer hemisphere the intensity in this sector is greatly reduced compared to the values for winter conditions.

4. Event studies of magnetic storms confirm the connection between anti-sunward auroral currents and the asymmetric storm-time disturbance signal. From the event studied we see that this claim holds for the total net current. But the actual current track can change during a storm several times between dawn and dusk sides in the two hemispheres. Most responsible for the preferred path is the prevailing season but also the IMF $By$ orientation.

5. Ground-based observations of the asymmetric disturbance signals confirm the seasonal dependence of larger values in the winter hemisphere. The horizontal disturbance component at the stations follows reasonably well the expected cosine-dependence with dipole latitude in the winter hemisphere. While for summer conditions a much faster reduction with latitude is observed. This hemispheric dependence implies a dominant role of the FAC magnetic fields for the asymmetric disturbance signals on ground.

6. We propose a 3D current system that connects during storm-times field-aligned currents with the anti-sunward high-latitude currents around noon and midnight and closes the loops through the magnetopause on the dawn and dusk flanks. We do not find evidence for a connection of this circuit with the ring current.

For confirming our claims about the large-scale current system causing the asymmetric storm-time magnetic disturbances more observations in the outer magnetosphere should be studied.

**Acknowledgements** The authors thank Ingo Michaelis for his great effort in improving the CHAMP Level 3 magnetic field data. The CHAMP mission was sponsored by the Space Agency of the German Aerospace Center (DLR) through funds of the Federal Ministry of Economics and Technology. The CHAMP magnetic field data (product identifier: CH-ME-3-MAG) are available at ftp://magftp.gfz-potsdam.de/CHAMP/L3_DATA. The OMNI data is available at ftp://spdf.gsfc.nasa.gov/pub/data/omni/high_res_omni/. The SMR data are downloaded from http://supermag.jhuapl.edu/indices/. The ground observations of magnetic



field are available at www.intermagnet.org. The work of Yun-Liang Zhou is supported by the
National Key R&D Program of China (No. 2018YFC1407303).

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





**Figures**


**Figure 1.** Schematic drawing of the net current determination approach at auroral latitudes
separately for dawn and dusk local time sectors. The unsampled virtual return paths are
shown as dashed lines.

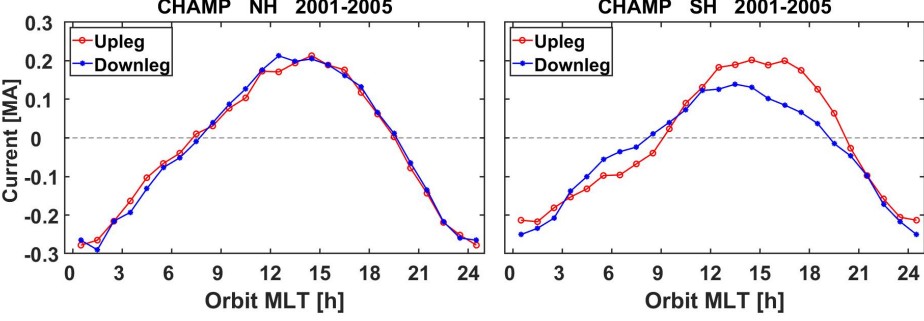

**Figure 2.** Local time dependence of auroral net currents separately for results from
upleg and downleg passes. Best matches, shown here, are obtained when the
contributions from the paths B-C and H-E are neglected.





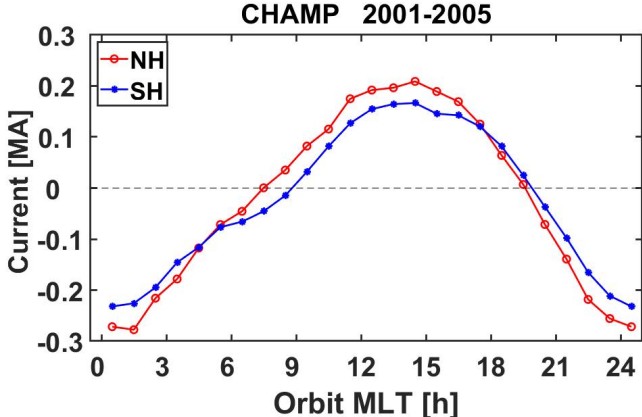

**Figure 3.** Local time dependence of mean auroral net currents; comparison between the two hemispheres.

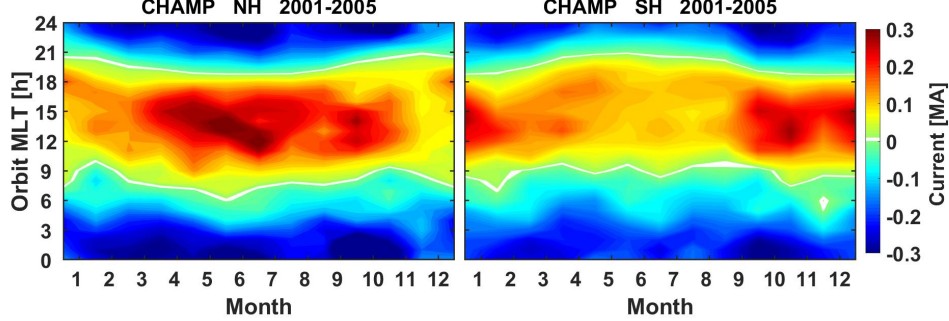

**Figure 4.** Distribution of mean eastward net currents in local time versus Month of Year frames. Noon-time currents are strongest during local summer seasons.



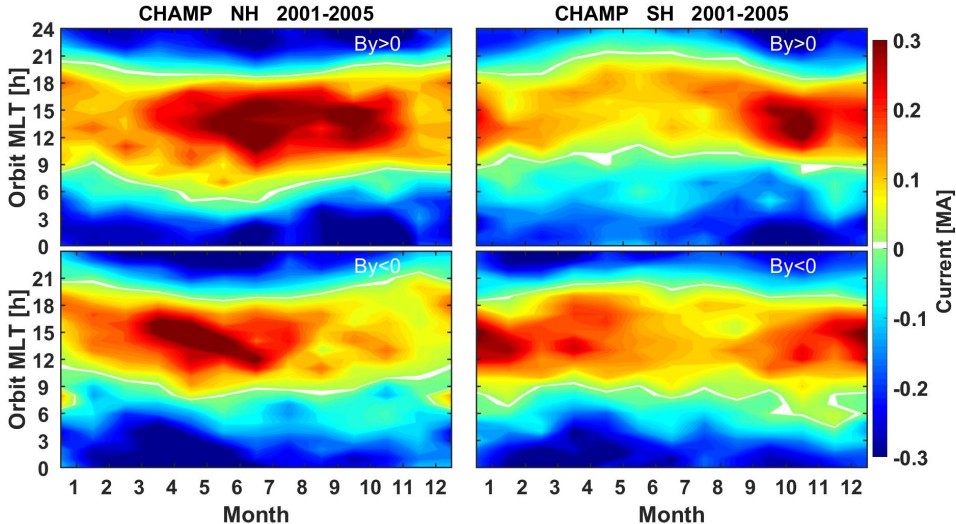

**Figure 5.** Distribution of mean eastward net currents in local time versus Month of Year frames separately for positive and negative IMF *By* conditions. DBY currents have the same direction as the cross-polar cap currents for *By* > 0 in the northern and *By* < 0 in the southern hemisphere.

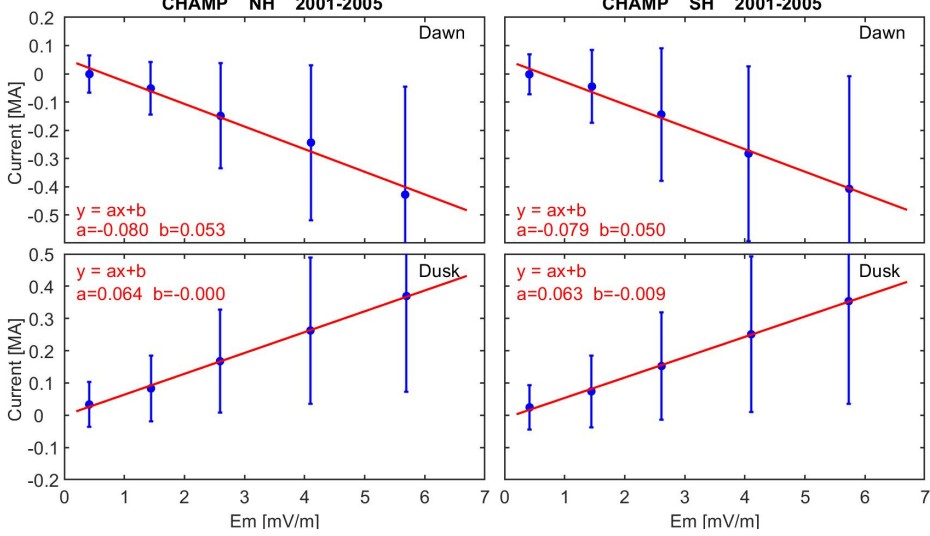

**Figure 6.** The $E_m$ dependence of net currents on the dawn and dusk sides, separately for the Northern (left) and Southern (right) hemispheres. The solid dots with vertical bars indicate the mean values and standard deviation of the net eastward current for five levels of $E_m$. Parameters of the linear fits (red lines) are listed in the top left corner of each frame.




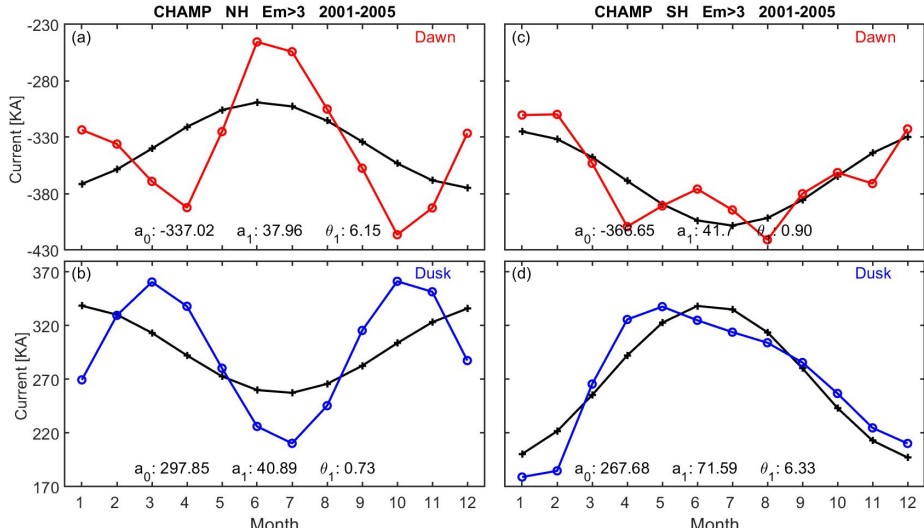

**Figure 7.** The seasonal variation of eastward net currents. Presented are dawnside (top) and
duskside (bottom) currents derived from high-latitude passes over the Northern (left) and
Southern (right) Hemispheres. Black curves are sinusoidal fits to the observations. In each
panel the constant term, $a_0$, annual amplitude, $a_1$ (both in kA) and the phases the peaks, $\theta 1$,
(in month) are listed.






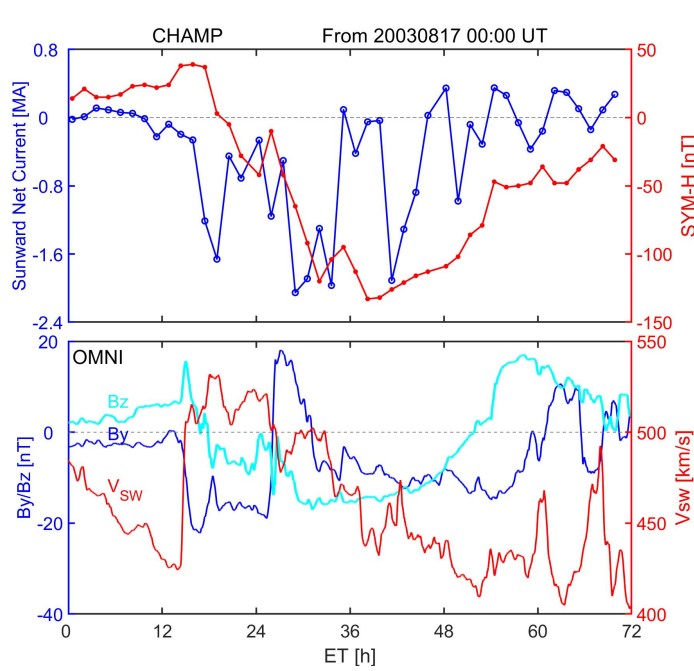

**Figure 8.** (*bottom*) Solar wind velocity and interplanetary magnetic field components (GSM)
variations for the storm starting on 17 August 2003. (top) The SYM-H index evolution during
the storm and the total anti-sunward net current are shown for comparison.




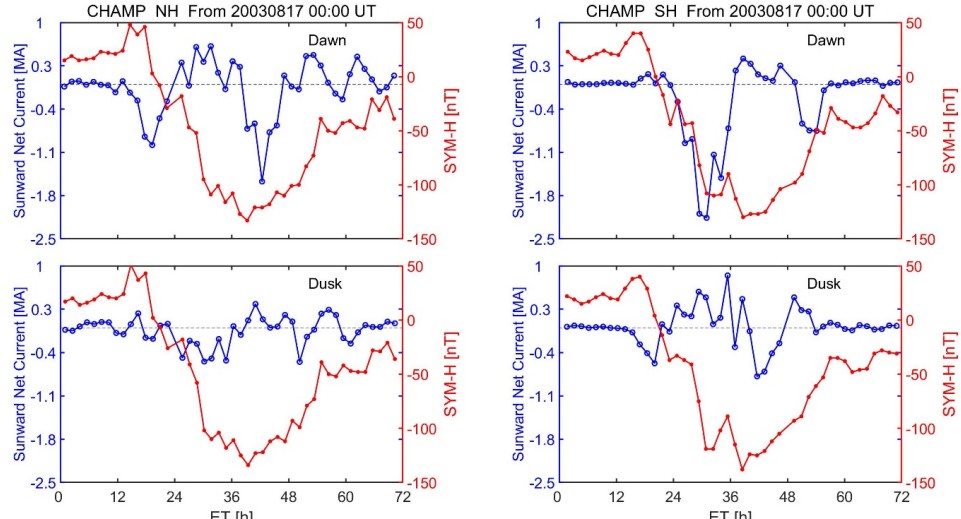

**Figure 9**. Temporal evolutions of the SYM-H index and the net currents separately for both hemispheres and for dawn and dusk sides during the storm 17-20 August 2003. Magnetic local time ranges are, NH dawn passes: 06-09 MLT, NH dusk passes: 17-21 MLT, SH dawn passes: 03-10 MLT, SH dusk passes: 17-23 MLT.


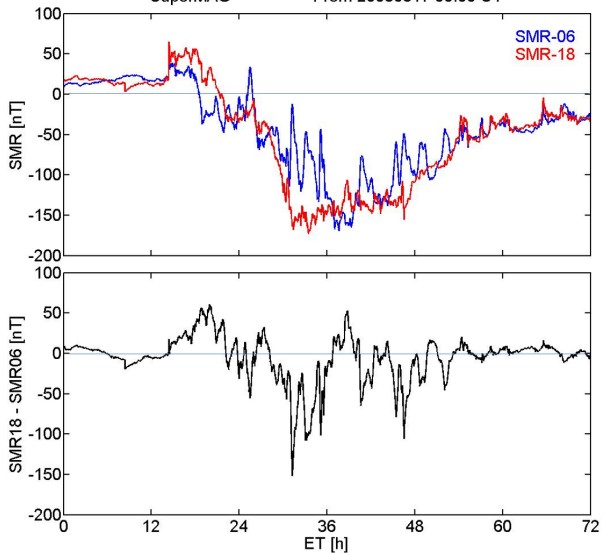

**Figure 10.** (top) Temporal evolution of SMR storm-time index from the 06 and 18 local time sectors. (bottom) Differences between the two time sectors (SMR-06 – SMR-18).




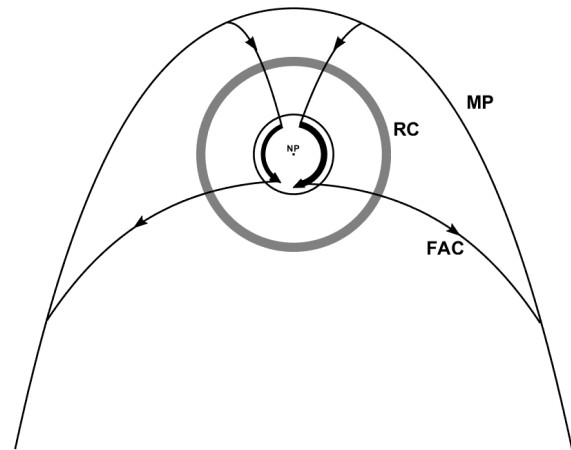

**Figure 11.** Schematic drawing of the suggested current circuits closing the high-latitude anti-
sunward currents. Field-aligned currents rout the net from the polar cap boundary on the
nightside to magnetopause on the dawn and dusk flanks. Here the currents flow sunward and
return back from the dayside magnetopause to the cusp regions in both hemispheres.





**Table 1.** The $E_m$ dependence of the net eastward currents during June and December solstice months for both the dawn and dusk sides.

| Season | Local time sector | Northern Hemis. | | Southern Hemis. | |
|---|---|---|---|---|---|
| | | Slope ($10^6$Am/V) | Inters. (kA) | Slope ($10^6$Am/V) | Inters. (kA) |
| Months: 05-08 | Dawn | -78 | 83 | -75 | 44 |
| | Dusk | 49 | 16 | 73 | -18 |
| Months: 11-02 | Dawn | -69 | 26 | -80 | 82 |
| | Dusk | 67 | 8 | 29 | 16 |

**Table 2.** The mean deflections of H component (in nT) at five observatories for different seasons during active times (Kp>=6)

| Station | Local time | June | December | equinoxes |
|---|---|---|---|---|
| AQU | Dawn | -21.8 | -97.5 | -46.3 |
| | Dusk | -71.3 | -146.8 | -88.3 |
| HER | Dawn | -35.3 | -77.2 | -52.0 |
| | Dusk | -104.8 | -122.8 | -106.0 |
| BNG | Dawn | -50.6 | -124.3 | -59.4 |
| | Dusk | -136.2 | -205.9 | -140.4 |
| TAM | Dawn | -38.7 | -109.7 | -66.4 |
| | Dusk | -112.2 | -184.9 | -127.0 |
| WNG | Dawn | -43.9 | -124.2 | -46.3 |
| | Dusk | 6.5 | -62.6 | -21.5 |





**Table 3.** Ratio of mean disturbance deflection at a given observatory with respect to the value
at the equator, separately for the three Lloyd seasons.

| Station | DLat | Local time | June | December | Equinoxes |
|---------|------|-----------|------|----------|-----------|
| BNG | 4.36° | Dawn | 1 | 1 | 1 |
| | | Dusk | 1 | 1 | 1 |
| TAM | 24.81° | Dawn | 0.76 | 0.88 | 1.13 |
| | | Dusk | 0.82 | 0.9 | 0.91 |
| HER | -33.86° | Dawn | 0.69 | 0.62 | 0.88 |
| | | Dusk | 0.77 | 0.6 | 0.76 |
| AQU | 42.45° | Dawn | 0.43 | 0.79 | 0.79 |
| | | Dusk | 0.52 | 0.71 | 0.63 |

**Table 4.** Ratio of mean disturbance deflection at a given observatory with respect to the value
at the equator, separately for the local seasonal conditions. Values can be compared with the
latitude-dependent cosine law.

| Station | DLat | Cosine law | Local summer | | Local winter | |
|---------|------|-----------|------|------|------|------|
| | | | Dusk | Dawn | Dusk | Dawn |
| BNG | 4.36° | 1 | 1 | 1.1 | 1 | 1 |
| TAM | 24.81° | 0.91 | 0.82 | 0.84 | 0.9 | 0.88 |
| HER | -33.86° | 0.83 | 0.6 | 0.62 | 0.77 | 0.76 |
| AQU | 42.45° | 0.75 | 0.52 | 0.47 | 0.71 | 0.79 |
