# Peer review of "Relation between the asymmetric ring current effect and the anti-sunward auroral currents, as deduced from CHAMP observations 3 4 Hermann Lühr1) and Yun-Liang Zhou2) 5 6 1) GFZ, German Research Centre for Geosciences, Section 2.3, Geomagnetism, 14473 Potsdam, 7 Germany. 8"

_Annales Geophysicae, 2020_

## Referee Comment (RC1) · Anonymous Referee #1 · 10 Mar 2020

Referee's report on "Relation between the asymmetric ring current effect and the anti-sunward auroral currents, as deduced from CHAMP observations" by Lühr and Zhou (MS#angeo-2020-3)

This manuscript studies average characteristics of anti-sunward net currents flowing in the high-latitude ionosphere from statistical analysis of the magnetic field data obtained by the CHAMP satellite. The anti-sunward currents increase as the coupling function between the solar wind and the magnetosphere increases. They are twice larger in the winter hemisphere than in the summer hemisphere. The main phase of

a magnetic storm is a favorable condition for development of the current. It was proposed that the ionospheric anti-sunward currents do not connect to the ring current but make a circuit with currents flowing along the magnetic field and at the dawn/dusk flank magnetopause. The data analysis is sound and the results are very clear. However, previous studies reporting the similar results are completely ignored. The manuscript should refer to these studies and discuss their new findings. Also, there are some points to be clarified. The reviewer thinks that the manuscript is worth publishing in Annales Geophysicae after it is revised according to the following comments.

1. Similar previous studies

The anti-sunward net currents have been studied in detail by the following papers. These studies should be referred to in the introduction. It should be also discussed how the present results are similar to/different from these studies. Iyemori (1990), JGG, doi:10.5636/jgg.42.1249.

Iyemori (2000), AGU Monograph #118, doi:10.1029/GM118p0331.

Nakano et al. (2002), JGR, doi:10.1029/2001JA900177.

Yamashita et al. (2002), JGR, doi:10.1029/2001JA900160.

Nakano and Iyemori (2005), JGR, doi:10.1029/2004JA010737.

2. Lines 207–226, Figures 4 and 5.

These sentences and figures do not focus on the sunward/anti-sunward net ionospheric currents and will confuse readers. The referee suggests omitting these parts.

3. Tables.

(a) There are four tables, each of which contains a lot of numbers. Although Tables 2–4 include important results, it is very difficult to understand what they show. With these tables, readers cannot follow section 6. These data should be displayed in figures (instead of deleting Figures 4 and 5 as suggested in comment 2).

(b) In Tables 3 and 4, some numbers do not match, although they are expected to be the same. For example, Hermanus in December has 0.62 and 0.6 in Table 3, but Hermanus in local winter has 0.76 and 0.77 in Table 4 (other stations have the identical values). Please confirm.

4. Lines 451–479.

These hanging paragraphs should be moved to a new subsection, probably, section 7.1 and the following subsections being renumbered.

5. Typos.

Line 41. closing –> closes

Lines 383. UT –> LT

Line 541 week –> weak

---

## Referee Comment (RC2) · Anonymous Referee #2 · 2 Apr 2020

This paper by Luehr and Zhou, is a reworking of champ data to look into the ground signatures of the asymmetric ring current in relation to the higher latitude auroral currents. They determine a number of aspects of current closure relating to storm time conditions (defined by Em) which are indicated by event and statistical analysis. The results will be of interest to the community. It is a clear account and presents a convincing statistical analysis. I recommend publication but have the following minor comments the authors may wish to consider. These are not critical on publication.

1. I would suggest the authors clarify better their meaning of the terms 'summer hemisphere' and 'winter hemisphere' in the abstract. The term is clear I the discussion but not perhaps when reading the abstract for the first time.

2. I wonder if Figure 1 can be made a little clearer. It is hard to grasp the first time.

3. The discussion of effects hinges on calculation of the total current. I realised this is discussed in detail in a previous paper, but since further assumptions have to be made, perhaps some indication on the possible error (e.g. missed current), depending on conditions and sampling, could be added.

---

## Author Comment (AC1) · 14 Apr 2020

Responses to the referees' reports on (MS#angeo-2020-3) "Relation between the asymmetric ring current effect and the anti-sunward auroral currents, as deduced from CHAMP observations"
by Lühr and Zhou

We would like to thank the two referees for their effort in carefully reading the manuscript and for making constructive comments. We are pleased that both of them regard the study as relevant and ask only for minor revisions. All their comments have been considered seriously and appropriate changes have been made in the revised manuscript. We are convinced that the paper has gained significantly by the revision. For the convenience of the referees, we first repeat below their comments and then add our responses in blue text. Major revisions in the manuscript are highlighted in bold face.

**Referee #1**

This manuscript studies average characteristics of anti-sunward net currents flowing in the high-latitude ionosphere (…)
The data analysis is sound and the results are very clear. However, previous studies reporting the similar results are completely ignored. The manuscript should refer to these studies and discuss their new findings. Also, there are some points to be clarified. The reviewer thinks that the manuscript is worth publishing in Annales Geophysicae after it is revised according to the following comments.

1. Similar previous studies
The anti-sunward net currents have been studied in detail by the following papers. These studies should be referred to in the introduction. It should be also discussed how the present results are similar to/different from these studies.
Iyemori (1990), JGG, doi:10.5636/jgg.42.1249.
Iyemori (2000), AGU Monograph #118, doi:10.1029/GM118p0331.
Nakano et al. (2002), JGR, doi:10.1029/2001JA900177.
Yamashita et al. (2002), JGR, doi:10.1029/2001JA900160.
Nakano and Iyemori (2005), JGR, doi:10.1029/2004JA010737.

Thank you for making us aware of the additional works of Japanese scientists on anti-sunward current studies. However, they are only partly relevant for this paper. All of the listed works make use of the magnetic azimuthal, *By*, component for estimating FACs and related anti-sunward currents. This approach depends on important assumptions and can only provide qualitative relations. Conversely, our ring integral of the along-track component is a more straight-forward approach that return quantitative values for the net current passing the polar region.
Even though, we now have made reference to these papers in the Introduction and Discussion sections.

2. Lines 207–226, Figures 4 and 5.
These sentences and figures do not focus on the sunward/anti-sunward net ionospheric currents and will confuse readers. The referee suggests omitting these parts.

We only partly agree with the reviewer's opinion. The reader first has to be introduced in the full distribution auroral net currents. It has to be made clear that the dawn to dusk net currents across the polar cap are dominating the distribution (Fig. 4). The anti-sunward component, of interest here, are just a secondary constituent.
For these arguments we prefer to keep Figure 4, but drop Figure 5 and the related text.

3. Tables.
(a) There are four tables, each of which contains a lot of numbers. Although Tables 2–4 include important results, it is very difficult to understand what they show. With these tables, readers cannot follow section 6. These data should be displayed in figures (instead of deleting Figures 4 and 5 as suggested in comment 2).

We largely followed the suggestion and significantly revised and improved Section 6 about the ground-based observations. Now the new Figure 10 displays the mean H component deflections on the dawn and dusk sides during disturbed periods at the 5 considered observatories separately for the seasons. Except for Wingst quite consistent results emerge. By reanalyzing the ground-based data, we have put more emphasis on determining the quiet-time backgrounds and removed spikes and jumps in the data. Numerical values for the mean dawn and dusk field values are listed in Table 2 and shown in Figure 11.
The new Table 3 lists the resulting dawn/dusk differences at the observatories separately for the seasons. The mean levels of magnetic activity, listed in Table 4, are needed for a proper interpretation of the derived asymmetries.
Overall, we are convinced that the manuscript gained significantly from the revision of the ground-based observations.

(b) In Tables 3 and 4, some numbers do not match, although they are expected to be the same. For example, Hermanus in December has 0.62 and 0.6 in Table 3, but Hermanus in local winter has 0.76 and 0.77 in Table 4 (other stations have the identical values). Please confirm.

We are afraid, this is a misunderstanding. In the old Table 3 the ratios had been sorted by global seasons: June, Dec. etc. While in Table 4 values had been sorted by local seasons: summer, winter, etc. But this is of no concern any more with the new tables in the revised manuscript.

4. Lines 451–479.
These hanging paragraphs should be moved to a new subsection, probably, section 7.1 and the following subsections being renumbered.

We followed the suggestion and added a new subsection heading:
7.1 Dependence on season and solar wind input

5. Typos.
Line 41. closing –> closes
Has been corrected, thank you

Lines 383. UT –> LT

Here the time in UT is correct for representing the dawn and dusk observations. The observatories are located between 0° and 30° longitude. This is now mentioned also in the text (lines 402-402).

Line 541 week –> weak
Has been corrected, thank you
* * *
**Referee #2**

This paper by Luehr and Zhou, is a reworking of champ data to look into the ground signatures of the asymmetric ring current in relation to the higher latitude auroral currents. They determine a number of aspects of current closure relating to storm time conditions (defined by Em) which are indicated by event and statistical analysis. The results will be of interest to the community. It is a clear account and presents a convincing statistical analysis. I recommend publication but have the following minor comments the authors may wish to consider. These are not critical on publication.

1. I would suggest the authors clarify better their meaning of the terms 'summer hemisphere' and 'winter hemisphere' in the abstract. The term is clear in the discussion but not perhaps when reading the abstract for the first time.

Now we make it clearer in the Abstract that we are comparing currents flowing through the polar regions in the summer and winter hemispheres.

2. I wonder if Figure 1 can be made a little clearer. It is hard to grasp the first time.

We have tried to make our schematic drawing of integration approach in Fig. 1 a little clearer. Now the directions of integration are indicated by arrows in the two loops. Also, the caption gives a more detailed description.

3. The discussion of effects hinges on calculation of the total current. I realised this is discussed in detail in a previous paper, but since further assumptions have to be made, perhaps some indication on the possible error (e.g. missed current), depending on conditions and sampling, could be added.

Uncertainties involved in our approach of determining net currents across the polar region are now mentioned at the end of Section 2 (lines 204ff). The additional assumption in this work is the neglection of contributions from the central vertical path elements to the current estimate. Any deviation from that assumption will not change the resulting amount of net current passing the polar region, but it will just affect it partitioning between the dawn and dusk sides.
Furthermore, we have added uncertainty bars to the mean annual variations of net currents in the new Fig. 6 (line 263).

---

## Author Response (AR2)

Response to Referee #1's report on "Relation between the asymmetric ring current effect and the anti-sunward auroral currents, as deduced from CHAMP observations" by Lühr and Zhou (MS#angeo-2020-3R)

The manuscript is revised according to the comments from the referees and is much improved. However, as stated below, there are a few points to be further considered before publishing in Annales Geophysicae.

We thank the referee for his/her effort in reviewing again our manuscript about asymmetric ring current and anti-sunward currents. We are pleased to note that only minor revisions are required for making it acceptable for publication. All the comments have been seriously considered, and in a point-by-point response we explain our motivation for the chosen phrases and presentations. Corresponding changes have been made in the manuscript, which make things clearer for the reader. Below, we first repeat the comments and then add our responses in blue text. Major changes in the manuscript are highlighted in bold face.

1. The revised manuscript refers to the previous papers studying the anti-sunward FAC current. Thus, reply is OK. But, the referee does not agree with the sentence "Conversely, our ring integral of the along-track component is a more straight-forward approach that return quantitative values for the net current passing the polar region.", because the method of authors includes a lot of assumptions to estimate the contributions from the unsampled parts and to calculate the ring integral.

Our chosen wording in the previous response to the referee's comments may not have been polite enough. But in the manuscript, no such decisive ranking of analysis techniques is expressed. We agree that a number of assumptions are made in our approach for calculating the ring integral from unsampled path elements. The resulting net currents, however, can be verified, at least in a statistical sense, by tracing them back to full-orbit integration, that does not require special assumptions. This validation chain is now described more clearly in lines 223ff, see also the response below.

2. Figure 4 is fundamentally the same as Figure 6 of paper by Zhou and Lühr [2017]. This figure is only stated and discussed in a very short paragraph (Lines 220–224). Thus, the referee is still uncomfortable in section 3. Either Figure 3 or Figure 4 is enough to demonstrate the full distribution of the net currents.

It is just the intension of Figure 4 to show the compatibility with Figure 6 in Zhou and Lühr [2017]. This is part of our line of arguments that the presented results are reliable. The average current density distribution presented here, although derived under more assumptions, agrees very well with that of Zhou and Lühr [2017], just the amplitudes are smaller by a factor of 2. This is expected since the dawn and dusk sectors are now sampled separately. In the Zhou and Lühr [2017] we have validated the separate results for the two hemispheres against the full-orbit net current results, which require practically no assumptions. Based on this chain of evidences we are quite convinced that the mean net current values presented here are reliable. For all these reasons we want to keep Figure 4, and have provided in lines 223ff the rational for it.

3. (a) Some tables are converted into figures, which greatly improve section 6. It becomes easier to understand the results of ground observations. However, Table 2 is yet something

redundant. If the authors want to show the numbers, please indicate in each panel of Figure 10.

We agree with the Referee that it is in principle possible to get mean magnetic field values from Figure 10. However, since the values listed in Table 2 are the basis of all subsequent calculations and considerations, we think, it is helpful for the reader to have the proper numbers when trying to verify the presented results. For that reason, we prefer to keep Table 2. These arguments are now outlined in the text, see lines 408ff.

(b) Reply is OK.

4. Reply is fine.

5. Replies are fine.

R1. The revised manuscript includes significant parts of addition in Lines 400–434 and Lines 493–502. In these additions, there are sentences "As a consequence, we have to state, our ground-based observations are not sufficient to reveal a possible seasonal effect of the storm-time disturbance asymmetry." (Lines 430–432) and "However, our statistical study of recordings from a single European-African meridional chain is not sufficient to confirm the seasonal difference between hemispheres." (Lines 493–494). This is a negative answer to a question raised in Lines 376–377, that is, "In order to obtain more information on the net current seasonal effects in ground observations we analysed magnetic field data from a meridional chain of observatories.", indicating the analysis of the ground station data provides no useful information. The analysis result does not have any consequence in interpretation of the anti-sunward net current. Thus, the part of these ground stations (i.e., the latter part of section 6) and related figures and tables can be deleted.

The Referee is right in stating that the meridional chain of observatories is not sufficient for determining seasonal dependences. But there are a number of other quantities that can be derived from their data, e.g. the degree of storm-time disturbance asymmetry and its relation to the strength of anti-sunward net currents. For these reasons we prefer to keep the results of ground-based observations in the paper. For providing a better connection between space and ground-based results we have added some sentences in lines 439ff.

R2. In titles of sections 4 and 5, "anti-sunward net current" may be better to understand the contents of these sections.

Thank you for the advices. We have implemented them.